# The Affective Ising Model: A computational account of human affect dynamics

**Tim Loossens** *, **Merijn Mestdagh**, **Egon Dejonckheere**, **Peter Kuppens**, **Francis Tuerlinckx**, **Stijn Verdonck**

Department of Quantitative Psychology and Individual Differences, KU Leuven, Leuven, Belgium

* tim.loossens@kuleuven.be

**Data Availability Statement:** All data files used in the paper ('Clinical ESM.csv','Longitudinal.csv') are available from the OSF database (accession number: osf.io/k8q5n/).

## Abstract

The human affect system is responsible for producing the positive and negative feelings that color and guide our lives. At the same time, when disrupted, its workings lie at the basis of the occurrence of mood disorder. Understanding the functioning and dynamics of the affect system is therefore crucial to understand the feelings that people experience on a daily basis, their dynamics across time, and how they can become dysregulated in mood disorder. In this paper, a nonlinear stochastic model for the dynamics of positive and negative affect is proposed called the Affective Ising Model (AIM). It incorporates principles of statistical mechanics, is inspired by neurophysiological and behavioral evidence about auto-excitation and mutual inhibition of the positive and negative affect dimensions, and is intended to better explain empirical phenomena such as skewness, multimodality, and non-linear relations of positive and negative affect. The AIM is applied to two large experience sampling studies on the occurrence of positive and negative affect in daily life in both normality and mood disorder. It is examined to what extent the model is able to reproduce the aforementioned non-Gaussian features observed in the data, using two sightly different continuous-time vector autoregressive (VAR) models as benchmarks. The predictive performance of the models is also compared by means of leave-one-out cross-validation. The results indicate that the AIM is better at reproducing non-Gaussian features while their performance is comparable for strictly Gaussian features. The predictive performance of the AIM is also shown to be better for the majority of the affect time series. The potential and limitations of the AIM as a computational model approximating the workings of the human affect system are discussed.

## Author summary

Feelings color and guide our lives. Understanding their dynamics is a crucial step on the way to eventually understanding mood disorders such as depression. In this paper, we propose a model for the dynamics of positive and negative affect, called the Affective Ising Model (AIM). Starting from a neurobiologically inspired yet abstract microscopic representation of how affect is generated, the model predicts the presence of a number of nonlinear phenomena in the dynamics of positive and negative affect. These nonlinear phenomena include skewed distributions, bimodality (people's affect can fluctuate around

**Funding:** This research was supported by the Research Fund of KU Leuven (grant no. GOA/15/003) and FWO (grant G074219N). M.M. and S.V. are supported by the Fund of Scientic Research Flanders. The computational resources and services used in this work were provided by the VSC (Flemish Supercomputer Center), funded by the Research Foundation|Flanders (FWO) and the Flemish Government, department EWI. The funders had no role in study design, data collection and analysis, decision to publish or preparation of the manuscript.

**Competing interests:** The authors have declared that no competing interests exist.

one of two possible states) and a V-shaped relation between positive and negative affect. These nonlinear signature features have been empirically established, but have thus far not been integrated into a single computation model. The AIM can be used in the future to explain both normal and dysfunctional affect.

## Introduction

Emotions play a prominent role in our lives. They guide our actions, help us interact with others and our environment, and help us make life decisions. A hallmark of affective phenomena (phenomena related to emotions, moods and feelings) is that they are not static but fluctuate throughout time due to external events and internal regulatory demands. These dynamics are referred to as affect dynamics [1]. It is exactly this temporal nature that renders affect its adaptiveness. It enables us to respond to opportunities and environmental threats, and motivates us to cope with them accordingly.

Because of its importance the dynamics of affect have become the subject of intensive empirical study. A variety of methods and paradigms exist to probe the feeling component of affect across time, both inside and outside the lab. Experience Sampling Methods (ESM) are generally considered to be the golden standard to study affect dynamics in an ecological valid manner [2]: people are measured repeatedly throughout the day during several days, giving researchers a window into their affective experiences during their daily lives.

Collecting intensive longitudinal data through ESM is only the starting point. Relevant information concerning affect dynamics has to be extracted from the data. Currently there are two major approaches of information extraction: calculating summary statistics and a model-based approach. The summary statistics approach relies on measures such as the autocorrelation [3] or the root-mean-square successive differences (RMSSD; [4]), but also simpler measures such as the mean, the variance and the correlation (for a more extensive overview, see [5]). The summary statistics focus on a particular (dynamical) aspect of interest and have the advantage that they are easy to understand intuitively. However, they also have a number of disadvantages. Many of the summary statistics tap the same kind of information (e.g., serial dependence, variability, etc.) and tend to correlate, which makes it challenging to understand the precise relations between them. Furthermore, they do not provide a broader framework in which affect can be understood and studied [5].

The second approach is model-based, meaning that a statistical model is fitted to the data. The most popular models for this purpose are autoregressive models [6–12]. In these models, the measurement or set of measurements at a given time point are regressed onto the measurement or set of measurements at the previous time point. This is an autoregressive model of order 1. For a set of measurements, the model is a vector autoregressive model. In most applications concerning affect dynamics, the model is assumed to be linear. A major advantage of these model-based approaches is that they tie together a number of the previously mentioned summary statistics in a dynamical framework. For example, a univariate autoregressive model of order 1 has parameters for the mean, variance, and autocorrelation, and thereby incorporates all these affect dynamical summary statistics. Another important advantage is that a statistical model is falsifiable; it can be used to make predictions about reality which can subsequently be tested against observation.

Although the current model-based approach is encompassing because it binds together a number of summary statistics and is falsifiable, it also comes with two major disadvantages. First of all, it is hard to construct a theoretically meaningful model for affect dynamics at the

right level of complexity. For instance, overly simple autoregressive models often outperform more complex models [13]. It is nevertheless unlikely that these overly simple models underlie affect dynamics, and some additional complexity is likely to be more meaningful. A second disadvantage is that, because of their linear and Gaussian structure, they are not able to capture a number of distinct qualitative phenomena: skewness, non-linear relations, and multimodality. These features have nonetheless been observed in data on affect, as will be discussed below.

A first feature deviating from Gaussianity that has been empirically observed in affect data is skewness [14–16]. In [17] it is even argued that the skew is an essential feature of the data and is likely to be subject to individual differences.

A second feature is the observed non-linear relation between components of the affect system. Of particular importance for the present purpose are the two major dimensions of affective experiences: positive and negative affect (abbreviated as PA and NA respectively). These are known to display a non-linear relation that reveals itself as a curved, V-shaped pattern (see [18]). Such a relation has been observed at the nomothetic level (i.e., across individuals; [19]), and in the valence-arousal space (which is—assumed to be—a rotated version of the PA-NA space by an angle of 45˚) it expresses itself as a V-shape [20]. In other words, whenever the positive or negative valence of the feeling component of affect increases, so does the arousal.

Third, the distribution of PA and NA scores is generally thought to be concentrated around a single location (the homebase or emotional baseline; [21]). However, theories and models for regime switching have been proposed in the context of bipolar disorder [22] and borderline personality disorder [23] because it is believed that people suffering from such disorders can switch between distinct homebases. In [24], bimodalities were also reported in the frequency distribution of depressive symptoms in people diagnosed with major depressive disorder. The underlying idea is that some individuals have two (or more) homebases and that they switch between the different regions across time.

In order to present a unifying account of the various summary statistics and explain the nonlinear empirical phenomena, a computational model is needed. This entails a mathematically formulated theory that represents at some abstract level the computations carried out by the brain and body to generate affect [25] and that can account for the aforementioned observations. In this paper, we introduce such a novel computational account called the Affective Ising Model (AIM). It is inspired by the Ising model, a model that has also been applied in cognitive psychology to study, for instance, decision making (the Ising Decision Maker; [26]). The model has some nonlinear features and therefore provides a framework that is very different from the typical Gaussian framework. It will be applied to PA and NA data of two different studies to investigate to what extent the model is capable of incorporating the observed non-Gaussian features (skew, bimodality, V-shape), both in healthy samples and people diagnosed with affective disorder. Moreover, to relate the model to other modeling efforts in the field, we will examine to what extent the model is able to reproduce the values of three commonly encountered summary statistics: the autocorrelation, the PA-NA correlation, and the root-mean-square of successive differences (RMSSD). In order to benchmark the AIM, the data will also be analyzed with a standard continuous-time vector autoregressive model, and an additional model intermediate between this continuous-time vector autoregressive model and the AIM will additionally be used as a comparison. Besides examining the capabilities of the models to reproduce the specific features and summary statistics, their predictive performance is compared by means of leave-one-out cross-validation.

## Model description

**Basic principles and ideas.**    On a psychological level, a general distinction has been made between a positive appetitive (approach) and a negative aversive (avoidance) subsystem of

affect (e.g., [27–32]). Likewise, the structure of affective experiences is for a large part accounted for by a two-dimensional space [19, 33]. The same distinction is also upheld at the neurobiological level. Evidence has been found (mostly in animal research) that there are dedicated and divergent biological structures for processing positive and negative affect related information in the brain (see e.g., [34]). Most likely the biological structures dealing with positive and negative affect information consist of a large number of smaller elements (neurons, neurotransmitters, hormones, etc.) that collectively produce positive and negative feelings. Each of these smaller elements is a degree of freedom of the system and contributes to the overall response: the experienced feeling of positive (PA) and negative affect (NA).

The starting point of the Affective Ising Model (AIM) is to make abstraction of the numerous elements, disregarding their specific function and exact connections, but preserving the general architecture of positive and negative components and the system's effective degrees of freedom. The AIM is made up of a vast collection of interconnected, stochastic, information processing units that switch on and off throughout time. These binary information units represent the degrees of freedom on a rudimentary, abstract level. As shown in Fig 1, they are divided into two large compartments or *pools*. One pool processes information leading to positive affect and another pool handles information giving rise to negative affect. To mimic the neuronal, biochemical and psychological processes in the brain and body, the binary information processing units excite one another within the same pool and inhibit one another between

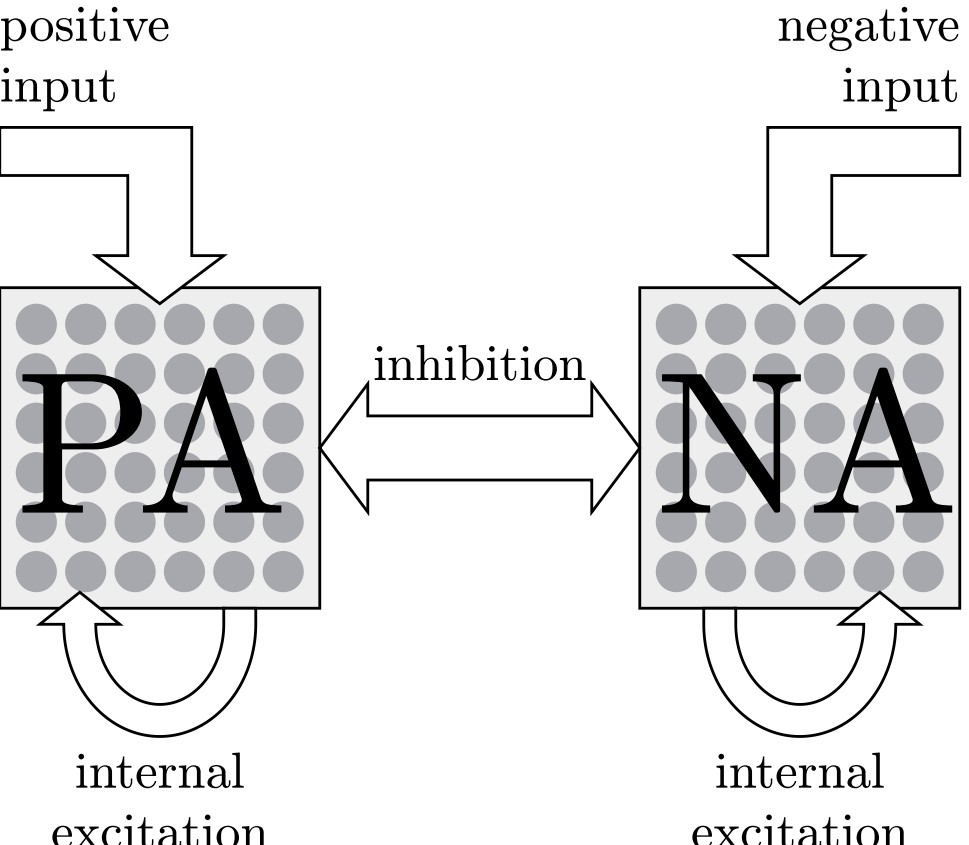

**Fig 1. Graphical representation of the AIM.** The binary information processing units (gray dots) are divided into two pools. One pool processes information regarding PA and the other processes information regarding NA. Information processing units of the same pool excite one another, whereas units from different pools inhibit one another. Positive and negative external stimuli can influence the information processing units of the PA and NA pool respectively.

pools. The information processing units can also be influenced by stimuli coming from outside of the affect system; positive and negative stimuli will respectively impact the units of the PA and NA pools. Because of the simplicity of the expression, the information units will be referred to as *binary neurons* in the remainder of the text (but they should not be equated with actual biological neurons).

The modeling approach taken in this paper has been successful in a gamut of research domains. It is an approach where information processing at a microscopic level is described abstractly, disregarding the specificity of the units themselves—whether they are spin magnetic moments, atoms, molecules or neurons. Making abstraction of the underlying processes facilitates the study of the emerging collective behavior of the system. This can be done using techniques from statistical mechanics. Models that rely on this procedure are often referred to as *Ising models* and they are used to help describe and explain a wide variety of phenomena, from phase transitions in magnets [35], to fluid dynamics [36], transport phenomena [37, 38], condensation transitions of gas molecules [39], the behavior of cell ensembles [40], bacterial chemotaxis [41], hysteresis in DNA compaction by Dps [42], and memory [43].

Thus, rather than providing a description of the affect system that is precise and accurate in all of its biological and psychological details, the AIM is meant to describe its emerging behavior, the experienced affect. The number of degrees of freedom of the affect system is very large. As a consequence it is unpractical, not to say unfeasible, to observe the underlying microscopic dynamics. This is comparable to a magnet because it is impossible to measure and directly model the large number of atomic spins. In spite of this lack of direct access to the microscopic level, however, the collective macroscopic behavior can be studied and related to observable phenomena, such as the magnetization of the material. So instead of considering the individual binary neurons (which in themselves do not necessarily represent distinct physical entities), the average activations of the pools in the AIM are assumed to be an immediate reflection of an individual's measurable affective experience. When many binary neurons of a pool are active, the individual will experience a strong sensation in correspondence to the strongly agitated pool. For instance, if a large proportion of PA pool neurons is active, the individual will experience a general positive sensation.

Referring to Marr's levels of analysis [44], the Affective Ising Model can be considered to be primarily situated at Marr's algorithmic level, describing how the affect system evolves and how it interacts with its environment. The model also makes assumptions about the implementational level, albeit rather abstractly. It assumes that the biophysiological structures underlying affect can be described by a multiple attractor network, an assumption common in computational neuroscience. The Affective Ising Model does, however, not speak to the the computational level. Affect is essential for an individual to evaluate its environment and context, and to determine how to respond, but the model does not specify why specific affective experiences are computed given specific conditions.

**Notation.** The AIM formally consists of two pools of stochastic binary neurons which are self-exciting and mutually inhibiting. The number $N_1$ of binary neurons in pool 1 (PA) is not necessarily equal to the number of binary neurons $N_2$ in pool 2 (NA). The average activations of the PA and NA pools will be denoted as $y_1$ ($0 \leq y_1 \leq 1$) and $y_2$ ($0 \leq y_2 \leq 1$), respectively. As the binary neurons switch on and off throughout time, the average activations also vary in time, giving rise to observable fluctuations in the affect state. Whenever required, the activations will be explicitly written as functions of time $t$: $y_1(t)$ and $y_2(t)$.

Self-excitation refers to the tendency of binary neurons belonging to the same pool to excite one another (like neurons firing together). The pairwise excitatory interaction strengths are assumed to be the same for all neurons of the same pool. On the collective system, these interactions result in positive feedback parameters $\Lambda_1$ and $\Lambda_2$ ($\Lambda_1, \Lambda_2 \geq 0$) controlling the self-

excitation of the average activations $y_1$ and $y_2$ respectively. Due to the mutual inhibition, binary neurons from different pools impede the activation of one another. These interaction strengths are also assumed to be equal for every pair of neurons. On the collective system, these interactions result in a negative interaction parameter $\Lambda_{12}$ ($\Lambda_{12} \leq 0$) that controls the inhibitory interaction between the two pools. Every binary neuron has a threshold to become active which is considered to be equal for every neuron belonging to the same pool. Across pools, the thresholds may differ. On the collective system, these result in two threshold parameters $\Theta_1$ and $\Theta_2$ ($\Theta_1, \Theta_2 \geq 0$) which indicate how easy or difficult it is to increase the average activity of the corresponding pool.

Finally, just like our affective experiences, the system of binary neurons responds to events from the outside world. Positively and negatively valenced events influence the activations of the neurons in pool 1 and pool 2. The impact of the outside world is accounted for by the AIM through the input strengths $B_1(t)$ and $B_2(t)$. These can be considered as the result of appraisal and other processes translating external events into input for the affect system. These inputs tend to either increase or decrease the average activity of the corresponding pool. If the environment is largely appraised as positive, $B_1$ will be much larger than $B_2$, thus exciting the PA more than the NA pool (whereupon the larger overall activation of the PA pool will suppress the average activity of the NA pool).

For data collected in daily life, one often does not have an idea of precise form of the stimulus functions $B_1(t)$ and $B_2(t)$ because information regarding external events is lacking or entirely absent. For simplicity, we assume in this paper that the influences from the environment are by and large a series of unrelated events of which some have more impact than others. This is a standard assumption that is implicitly present in many continuous-time stochastic models of affect (e.g., [21, 45, 46]). Absorbing such an input process into the model does not affect the model's properties (e.g., structure, dynamics, distributions, etc.). Hence, the model is still an AIM as described below. Therefore, we will not consider input further in the remainder of the text.

**Mathematical framework.** In the AIM, the affect state $\mathbf{y}(t) = (y_1(t), y_2(t))$ of an individual at time $t$ corresponds to a point in the two-dimensional PA-NA space. Because both $y_1$ and $y_2$ are bounded by 0 and 1, the affect space (PA-NA plane) corresponds to a unit square. For each individual, there are regions of the affect space that are visited more often, and regions that are rarely visited. This is similar to how some affective sensations are experienced more often by someone, while others are seldom experienced. This is reflected in a bivariate probability distribution $p(y_1, y_2)$ on the affect space. In the AIM, this distribution is described by

$$p(y_1, y_2) = \frac{e^{-\beta F(y_1, y_2)}}{\mathcal{Z}}, \qquad (1)$$

where $F(y_1, y_2)$ is a bivariate free energy function and $\mathcal{Z} = \int_0^1 \mathrm{d}y_1 \int_0^1 \mathrm{d}y_2 \, e^{-\beta F(y_1, y_2)}$ is a normalizing constant ensuring that $p(y_1, y_2)$ is a proper probability density function. The parameter $\beta$ is a remnant from statistical mechanics where it is identified with the inverse temperature. Within the framework of the AIM, $\beta$ is unidentified and therefore redundant (see [26]). It will be set to 1 in the remainder of this text. However, for the sake of completeness, $\beta$ will be kept in the equations.

The free energy function contains information pertaining to the structure of the system of binary neurons. In terms of the model parameters $\{\Lambda_1, \Lambda_2, \Lambda_{12}, \Theta_1, \Theta_2, N_1, N_2\}$, which pertain

to the configuration of the affect system itself (i.e., no inputs), the free energy is defined as

$$F(y_1, y_2) = \sum_{i=1}^{2}(-\Lambda_i \, y_i^2 + \Theta_i \, y_i) + \Lambda_{12} \, y_1 \, y_2$$

$$+\sum_{i=1}^{2} \frac{N_i}{\beta} \, (y_i \ln(y_i) + (1 - y_i) \ln(1 - y_i)).$$

(2)

If the affect system is influenced by (time-varying) external inputs, $B_1(t)$ and $B_2(t)$, the linear terms $\Theta_i \, y_i$ have to be substituted with $(\Theta_i - B_i(t))y_i$ (see [26]). In that case, both the internal configuration and the environment shape the free energy. We will come back to this in the discussion. However for now, we will work with a free energy that is fully determined by the internal configuration of the affect system.

With $\beta$ fixed to 1, the relation between free energy functions in Eq (2) and affect distributions in Eq (1) is one-to-one. Therefore, each individual has her own affect distribution that is determined by his unique free energy function. It essentially translates the microscopic degrees of freedom into a distribution over macroscopic affect states. A wide variety of affect distributions can be accommodated by the free energy function in Eq (2). A few examples are shown in Fig 2. The distributions are plotted as density maps for which darker regions correspond to areas where the affect state is more likely to be observed. In order to be realistic, the distributions shown are based on estimated parameters derived from empirical data (see below). Configurations that are typically encountered in data include unimodal and Gaussian-like distributions as in Fig 2(a), unimodal but skewed distributions as in Fig 2(b), bimodal distributions as in Fig 2(c), and distributions with a V-shaped relation as in Fig 2(d).

In Fig 3, an example of a free energy function is shown with its corresponding affect distribution underneath. It can be seen that the free energy function can literally be interpreted as an emotional 'landscape' on which the affect state roams. If we would measure the affect state of a specific individual many times across time, the observations would start forming his affect distribution from Eq (1) which is entirely determined by the shape of the landscape. The affect distribution from Eq (1), however, has also got a more probabilistic interpretation: if you were to measure a person's affect state once and you would wait for a sufficiently long time, then at the time of the next measurement, the distribution in Eq (1) describes the probability of observing the affect state in a specific region. It provides us with information about where the affect state is likely to be observed and where not.

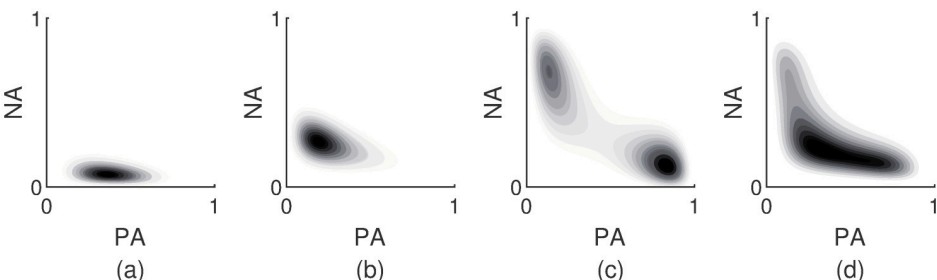

**Fig 2. Examples of particular affect distributions.** The examples are based on estimates derived from observed data. Panel (a) exhibits a unimodal distribution with Gaussian-like features. Panel (b) depicts a unimodal, skewed distribution. Panel (c) is an example of a bimodal distribution. In panel (d) a clear V-shaped relation between the PA and NA dimensions in visible.

The time between subsequent measurements is essential. Measurements close to each other in time are expected not to differ too much. In other words, the affect state does not get randomly sampled from the affect distribution each measurement time. Rather, the affect state wanders around more or less continuously on the free energy surface. The motion is not entirely continuous because of the inherent stochasticity of the microscopic system of binary neurons that is translated into random fluctuations of the macroscopic affect state. Besides these fluctuations, the affect state experiences a drift due to the sloping of the landscape so that the affect state is pulled downwards. Only through the random fluctuations can it move uphill. As can be seen in Fig 3, the densest region of the affect distribution coincides with the minimum of the free energy surface. Due to the downward pull, the affect state is more likely to

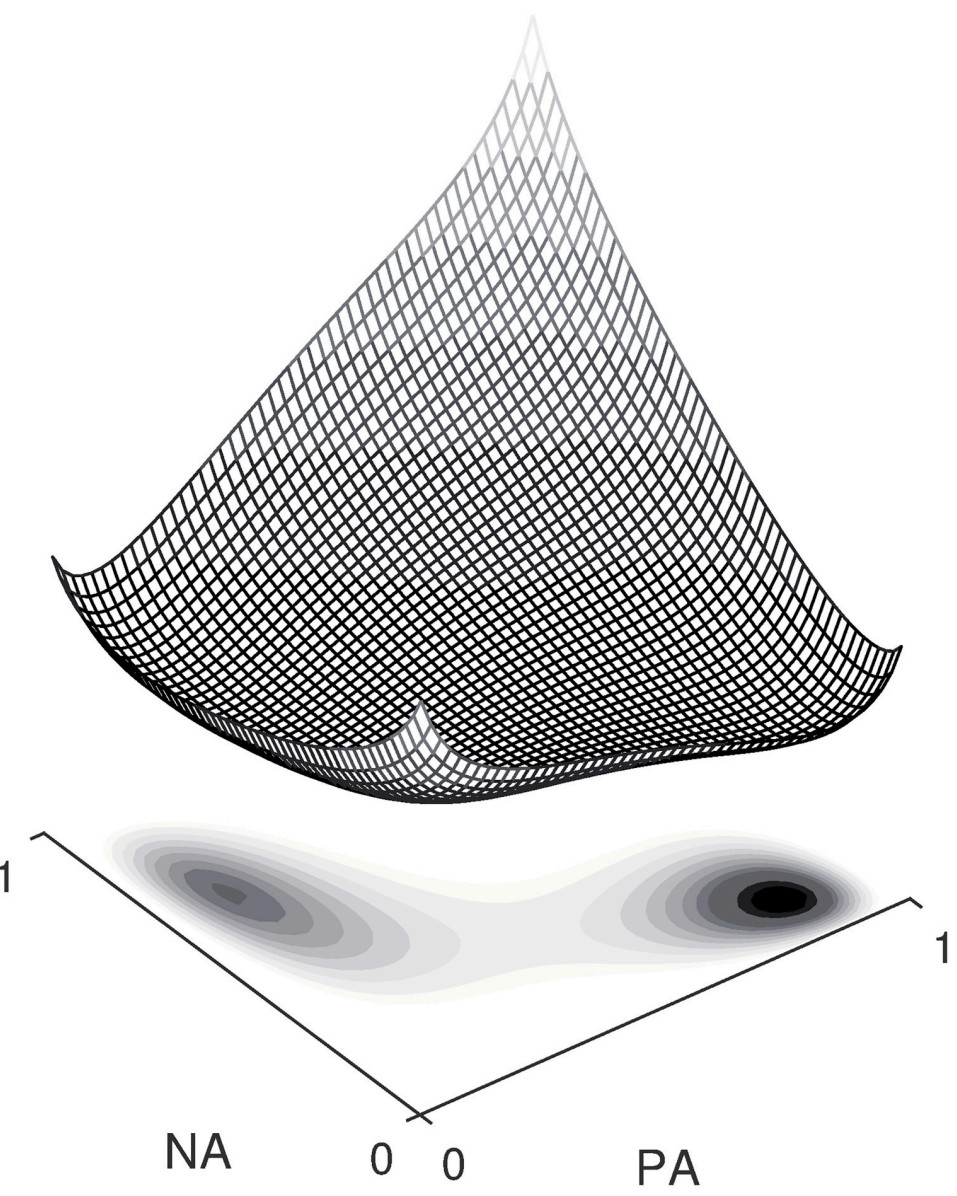

**Fig 3. Illustration of a free energy surface.** The corresponding affect distribution in Eq (1) is depicted as a density map beneath the surface. The darker, the more likely it is to observe the affect state in that region.

dwell in (local) minima of the surface. It however takes time for the affect state to move from one place to another. So after a measurement, the affect state will keep on moving randomly—more downhill than uphill—but as long as the waiting time until the next measurement is short, the affect state will not have moved far away from its previous location.

The movement of the affect state across the surface throughout time is determined by the dynamics of the underlying microscopic system (the continuous switching on and off of the binary neurons). These microscopic dynamics bring forth the dynamics of the macroscopic dynamics of the affect state described by the equations (for a derivation, see [26])

$$
\begin{aligned}
\mathrm{d}y_1(t) &= -\beta D \frac{\partial F(y_1(t), y_2(t))}{\partial y_1}\,\mathrm{d}t + \sqrt{2D}\,\mathrm{d}W_1(t),\\
\mathrm{d}y_2(t) &= -\beta D \frac{\partial F(y_1(t), y_2(t))}{\partial y_2}\,\mathrm{d}t + \sqrt{2D}\,\mathrm{d}W_2(t).
\end{aligned}
\tag{3}
$$

The first terms on the right hand side of Eq (3) are called drift terms. These terms are the cause of the downward pull. The second part is the diffusion part, responsible for the inherent randomness; $W_1(t)$ and $W_2(t)$ represent two uncorrelated Wiener processes. The parameter $D$ ($D > 0$) is another model parameter called the diffusion constant and determines the pace at which the affect state moves across the surface. Two individuals with exactly the same free energy surface but a different diffusion constant will have the same affect distribution as given by Eq (1) but will move around at different paces. The smaller $D$, the slower the affect state evolves.

So shortly after a measurement, the affect state is expected to be observed in the neighborhood of its previous location (the system of binary neurons has a certain inertia). By solving Eq (3), the conditional probability density can be obtained which tells you were the affect state is likely to be found and where not after a specific time interval. The longer you wait, the more the conditional probability density will converge towards the affect distribution from Eq (1). After a long time, the affect state can have reached any location covered by the free energy function and its exact location at the previous measurement is no longer informative for its current position. At that point, the conditional probability distribution does no longer change across time (it stabilizes) and the only information we have left lies in the shape of the landscape (i.e., Eq (1) is our only source of information to make an educated guess of where the affect state could be).

In Fig 4, a graphical illustration of a time evolution is given. In the four panels, the altitude lines of the free energy surface are drawn in red. Panel (a) depicts the conditional probability density shortly after a measurement. The probability distribution is sharply peaked around the location of the previous observation, which coincides with the center of the black dot. In panel (b), the conditional probability has smeared out a bit more and the densest region has shifted slightly towards the closest local minimum of the surface. This is the situation after waiting a while longer. In panel (c), we see that the affect state could have reached the other minimum by now. The waiting time is much longer than in panel (b). Still, the conditional probability density has not yet entirely relaxed towards the affect distribution given by Eq (1). The densest region is still located at the local minimum in the top left corner of the affect space. In panel (d), the conditional probability density has entirely relaxed towards the affect distribution from Eq (1). The affect state could have reached anywhere within the boundaries set by the free energy surface. All information pertaining to where the affect state started its journey is lost.

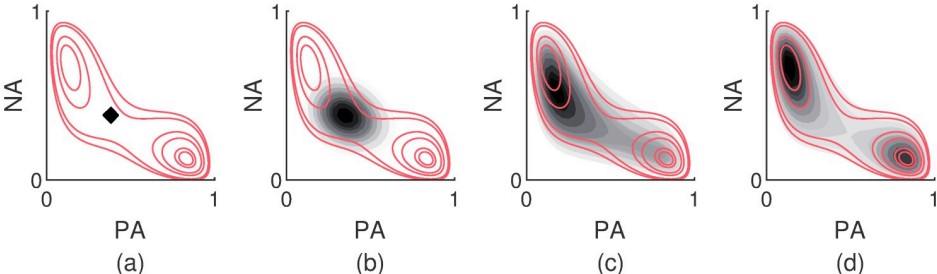

**Fig 4. Illustration of how the affect state evolves after a measurement.** The density maps in the different panels correspond to the conditional probability density which describes how far away the affect state could have moved since the last measurement. The red lines denote the altitude lines of the free energy surface. Panel (a) illustrates the situation shortly after the measurement. The black dot coincides with the position of the affect state at the last measurement. In panel (b), some time has passed since the last measurement. The conditional probability density has smeared out over the surface. Moreover, the central black dot that corresponds to the most likely region has shifted slightly towards the closest local minimum of the surface in the upper left corner. Panel (c) depicts the conditional probability density after waiting a long time. The affect state could have reached far away by now, but the conditional probability has not yet entirely relaxed to the distribution given by Eq (1). In Panel (d) the conditional probability density has entirely relaxed. All information pertaining to where the affect state was at the time of the last measurement is lost. The affect state could have reached anywhere in the affect space within the boundaries of the free energy surface.

## Results

### Explaining the nonlinear empirical features

In the introduction, we listed three empirically observed features in affect data that should be captured by a computational model of the affect system: skewness, the V-shaped relation between PA and NA, and bimodality. In Fig 2, some example AIM distributions were shown which exhibit these features.

To analyze to what extent the AIM is able to systematically capture the empirical features, we defined for each of them a statistic: the centralized third moment $m_3$ for skewness, a curvature index $\kappa$ for V-shape, and the bimodality coefficient $BC$ for bimodality (for more information on how these statistics were defined, see the Materials and methods section). These statistics were estimated for 685 experience sampling data sets (see Materials and methods for a description of the data).

Next, three models were fitted: (1) the AIM, (2) a standard continuous-time vector autoregressive model, the Ornstein-Uhlenbeck (OU) model [21, 45], and (3) a truncated variant of the OU model in which hard boundaries at the edges of the unit square are introduced. The third model is referred to as the bounded OU model. The OU model is only able to produce Gaussian distributions, while the bounded OU model is able generate truncated Gaussian distributions on the affect space. They served as benchmark models to compare with the AIM.

Using a parametric bootstrap (see e.g., [47]), distributions of replicated statistics were obtained for each of the models. This allowed us to evaluate to which extent the observed features can be captured by each of the models.

**Skewness.** The results of the parametric bootstrap of the standardized third moments $m_3$ are shown in Fig 5. In each of the panels, the median of the replicated standardized third moments $m_3^{rep}$ is set out against the observed third moments $m_3^{obs}$; they are depicted as darker dots. The replication based 90% confidence intervals are depicted as lighter bars. The first column corresponds to the results that have been obtained for the plain OU model. Similarly, the results shown in columns two and three have been obtained using the bounded OU model and the AIM, respectively. The first row corresponds to the PA dimension and the second row

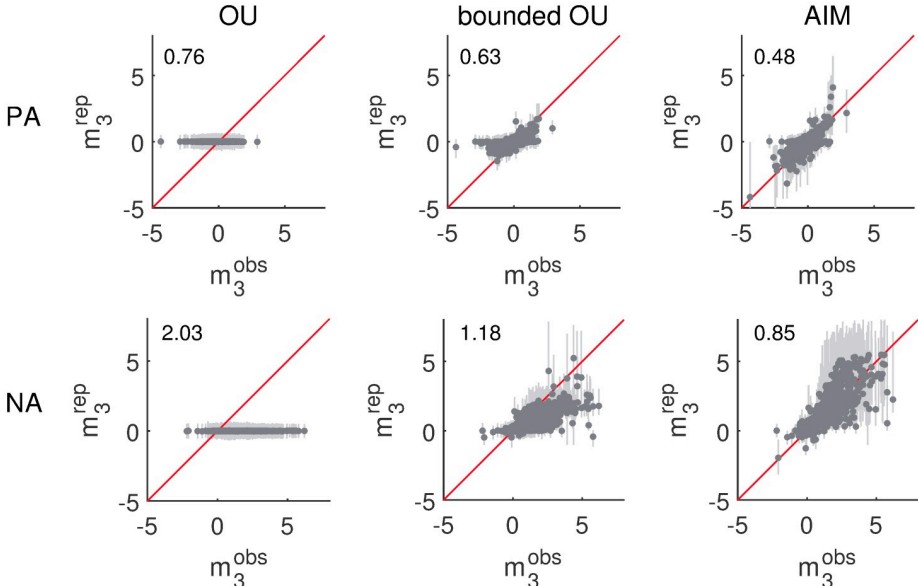

**Fig 5. Skewness results.** The darker dots denote the medians of the replicated standardized third moments $m_3^{rep}$ in function of the observed third moments $m_3^{obs}$ for the different data sets. The lighter bars correspond to the 90% confidence intervals based on the distribution of $m_3^{rep}$. The first, second and third columns respectively correspond to replications based on the plain OU model, the bounded OU model and the AIM. The first and second rows correspond to the PA and NA dimension respectively. The root-mean-square error between the median and observed third moments is indicated in the upper left corner of each panel.

corresponds to the NA dimension. The replicated standardized third moments $m_3^{rep}$ are supposed to be equal to the observed third moments $m_3^{obs}$. Whenever this is the case, the darker dot lies on the main diagonal drawn in red.

In the first row, we see that the plain OU model, being inherently Gaussian, is not able to replicate the observed skewness. Gaussian distributions have no skewness. For most of the data sets, the PA and NA scores are nevertheless skewed. The distribution of observed third moments $m_3^{obs}$ is centered around zero for the PA dimension, but the NA dimension in general has a more positive skew. The replicated third moments $m_3^{rep}$ obtained using the plain OU model, on the other hand, are small and always centered around zero; the observed trend is not replicated. If the medians of the replications are considered as estimates for the standardized third moments, the root-mean-square error is 0.76 and 2.03 for the plain OU model in the PA and NA dimension respectively.

Distributions produced by the bounded OU model are no longer Gaussian. These distributions originate from Gaussian distributions but are truncated at the edges of the unit square. As a consequence, these distributions are not necessarily, and will in general not be, symmetric. Therefore, the third moment of these distributions can be different from zero. As can be seen in the second column of Fig 5, the bounded OU model is to some extent able to reproduce the observed skewness in both the PA and the NA dimensions. However, it seems as if the model has got difficulty reproducing larger positive and negative skewness; the dots deviate more from the diagonal in these regions. Typically, the reproduced skewness is smaller in absolute value than the observed skewness. The root-mean-square error is 0.63 and 1.18 for the PA and NA dimension, respectively.

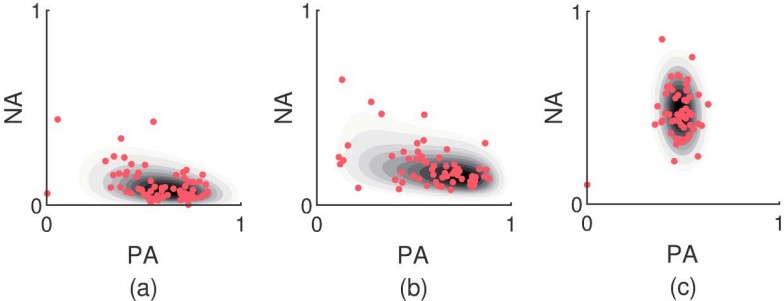

**Fig 6. Examples of skewed AIM distributions.** In panels (a) and (b) two fitted AIM distributions are shown which give rise to data patterns with a large skewness in the NA dimension (5.46 and 4.88 respectively). The underlying observed data patterns are shown as well. The distribution in panel (c) corresponds to an AIM distribution with little skewness.

In the third column (results for the AIM), we see that the darker dots more or less follow the red diagonal. There are no deflections for larger positive or negative skewness like for the bounded OU model. The AIM is able to generate large skew both in the positive and in the negative direction. The root-mean-square error of the median estimates is 0.48 and 0.85 for the PA and NA dimensions, respectively.

Panels (a) and (b) of Fig 6 exhibit two examples of AIM distributions with a large skewness in the NA dimension. The observed data patterns are demonstrated as well. The parametric bootstrap led to a median third moment of 5.46 and 4.88 for the two marginalized NA distributions respectively. In Panel (c) of Fig 6 an example is shown of a distribution with almost no skewness.

In conclusion, the empirically observed skewness is best captured by the AIM. The plain OU model cannot produce any skewness and although the bounded OU model can do so, it has got difficulty with generating heavily skewed distributions.

**V-shape.**    The results of the parametric bootstrap of the curvatures $\kappa$ are shown in Fig 7. The structure of the figure is analogous with Fig 5, but contrary to Fig 5 there are no separate figures for PA and NA.

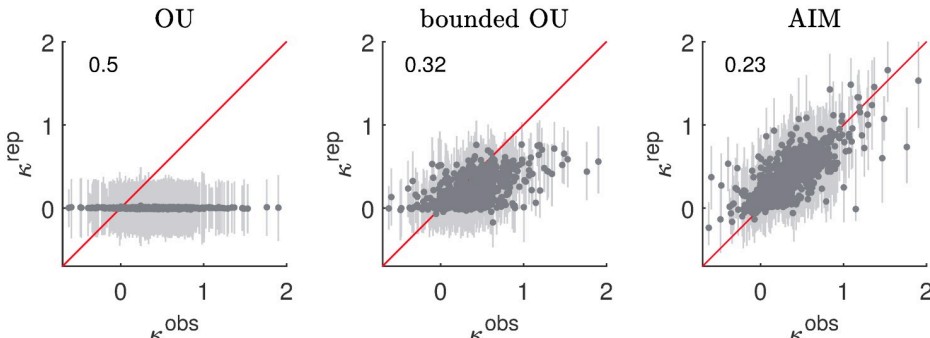

**Fig 7. V-shape results.** The darker dots denote the medians of the replicated curvatures $\kappa^{rep}$ as a function of the observed curvatures $\kappa^{obs}$ for the different data sets. The lighter bars correspond to the 90% confidence intervals based on the distribution of $\kappa^{rep}$. The first, second and third columns respectively correspond to replications based on the OU model, the bounded OU model and the AIM. The root-mean-square error between the median and observed curvatures is shown in the upper left corner of each panel.

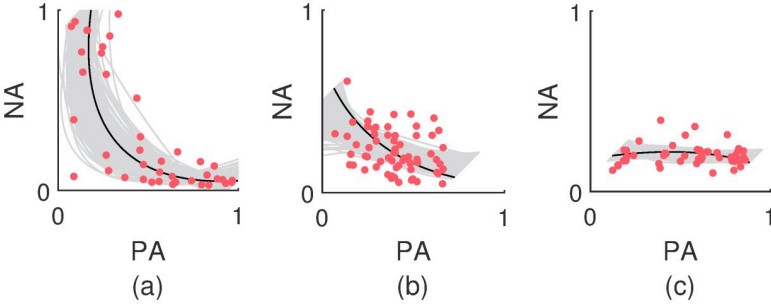

**Fig 8. Examples of V-shaped AIM distributions.** Three examples of data patterns and their associated least-squares parabolas. The lighter lines depict an AIM based parametric bootstrap of the least-squares parabolas. Panel (a) depicts a clear V-shaped data pattern that stretches out over the entire affect space. The data pattern in Panel (b) is more elliptic. The least-squares parabola shows little curvature and follows the major axis of the ellipse. The data pattern in Panel (c) is an example of a data pattern with an inverted least-squares parabola which has a negative curvature.

As can be seen in the left panel of Fig 7, the curvature produced by the plain OU is essentially zero (although there is some uncertainty because finite samples may show small curvatures). The middle panel of Fig 7 indicates that the bounded OU model can reproduce larger curvatures than the plain OU model. However, the reproduced curvatures are in general smaller than the observed curvatures. It is especially difficult for the bounded OU model to reproduce large curvatures (i.e., curvatures larger than 1). Considering the medians as estimates for the curvatures, then the root-mean-square error is 0.32. This is smaller than the root-mean-square error of 0.50 obtained for the plain OU model. As can be seen in the right panel of Fig 7, the AIM is capable of reproducing both small and large curvatures. The blue dots follow the diagonal; the root-mean-square error of the median curvatures is 0.23.

For the AIM, the sign of the interaction parameter $\Lambda_{12}$ determines whether there is inhibition ($\Lambda_{12} > 0$) or excitation ($\Lambda_{12} < 0$). Because inhibition is theoretically meaningful, a lower bound of zero was imposed on $\Lambda_{12}$. This raises the question what happens when the lower bound of zero on the interaction parameter $\Lambda_{12}$ is removed. It appears that mutual excitation occurs in about 5% of the cases and that in all these cases the interaction strength is very small. No strong evidence could be found for the existence of excitatory interactions between the PA and NA dimensions, which also explains why the AIM is capable of reproducing the observed curvatures in the data.

In Fig 8, three data patterns and their corresponding least-squares parabolas are depicted (see the Materials and methods section for more information on least-squares parabolas). The lighter lines correspond to an AIM based parametric bootstrap of the least-squares parabolas. Panel (a) corresponds to a clear V-shaped data pattern that stretches out over the entire affect space. In Panel (b), the data pattern is more in line with a normal distribution and as a result, the least-squares parabola shows little curvature. This kind of curvature can be reproduced by the (bounded) OU model too. The data pattern in Panel (c) does not correspond to a V-shape, but it is an example of a data pattern with an inverted least-squares parabola. Such negative curvatures occur in 10% of the cases.

In conclusion, V-shaped distributions cannot be produced by Gaussian models like the plain OU or bounded OU model. The bounded OU model is capable of producing some of the smaller empirically observed curvatures, but it does not compare to the AIM when it concerns larger curvatures. It is by virtue of the inhibition parameter $\Lambda_{12}$ that the AIM is able to create the V-shaped distributions like in Fig 8 and no statistical evidence could be found that the assumption of inhibition would be unreasonable.

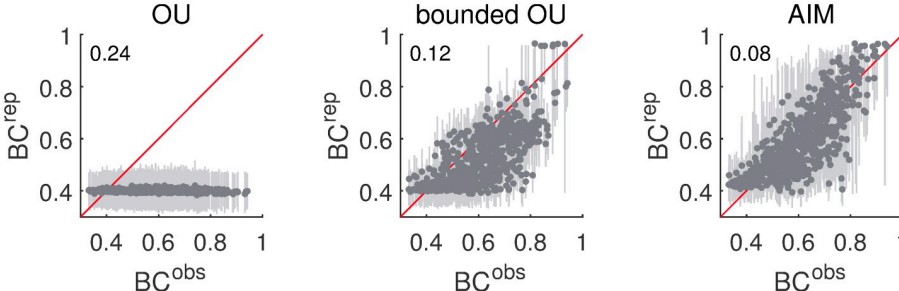

**Fig 9. Bimodality results.** The medians of the replicated bimodality coefficients $BC^{rep}$ set out against the observed bimodality coefficients $BC^{obs}$ for the different data sets. The lighter bars correspond to the 90% confidence intervals based on the distribution of $BC^{rep}$. The first, second and third columns respectively correspond to replications based on the plain OU model, the bounded OU model and the AIM. The root-mean-square error between the median and observed bimodality coefficients is indicated in the upper left corner of each panel.

**Bimodality.**  The results of the parametric bootstrap of the bimodality coefficients are shown in Fig 9. The structure and information content of the figure is the same as for Fig 5.

Observed bimodality coefficients $BC^{obs}$ range from 0.33 to 0.94. The average is equal to 0.61 which exceeds the threshold value $BC_{crit} \approx 0.5556$ (the $BC$ value of a uniform distribution). In total, 65.2% of the data sets give rise to a $BC$ value larger than the threshold. Nonetheless, the plain OU model produces affect distributions with $BC$ values close to 0.4 regardless of the observed bimodality coefficients. The root-mean-square error between the medians of the replicated bimodality and the observed bimodality coefficients is 0.25.

The bounded OU model, on the other hand, is able to generate larger $BC$ values. It is known that heavily skewed unimodal distributions can also give rise to large bimodality coefficients. As a result, not all of the distributions with larger $BC$ values will be bimodal. If the large $BC$ value is caused by heavy skew, the bounded OU model might be able to reproduce this value to some extent. However, the large density of dots below the diagonal indicates that the bounded OU model is not entirely able to reproduce the observed bimodality coefficients; the root-mean-square error between median and observed values is 0.12.

For the AIM the dots are more equally distributed about the diagonal. This model captures the whole spectrum of $BC$ values. The root-mean-square error between the median replicated $BC$ values and the observed values is 0.08.

The substantial number of $BC$ values larger than the threshold $BC_{crit}$ is serious evidence for bimodality. According to the results of the non-parametric bootstrap of the percentage of participants with a truly bimodal AIM distribution, on average 35% of the data sets have multiple modes. The 90% confidence interval is given by [32%; 37%]. In Panels (a) and (b) of Fig 10, two examples are shown of AIM distributions that consistently turn up bimodal. The underlying data patterns are also depicted. Fig 10(c), on the other hand, is an example of an AIM distribution that is consistently unimodal. Such data patterns can also be reproduced by the (bounded) OU model.

In conclusion, a large number of empirically obtained $BC$ values is larger than the threshold value that is expected for uniform distributions. The AIM is best at reproducing the bimodality coefficients, both larger and smaller ones. Unlike the OU model, the bounded OU model is also capable of generating data patterns with larger $BC$ values, but it generally underestimates the true observed values. This however does suggest that some of the larger $BC$ values are caused by heavy skew instead of bimodal patterns. According to the estimations of the AIM, about 35% of the data sets have multiple modes.

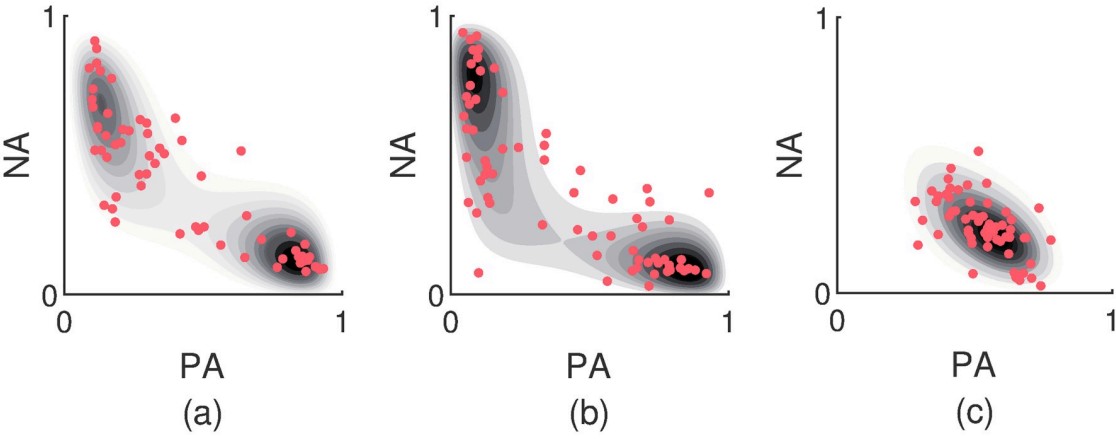

**Fig 10. Examples of bimodal AIM distributions.** Panels (a) and (b) demonstrate fitted AIM distributions that are consistently bimodal across non-parametric bootstraps. The underlying observed data patterns are shown as well. Panel (c) corresponds to a unimodal, Gaussian-like data pattern.

## Reproducing common summary statistics

The models were also evaluated regarding how well they are able to describe three of the more common summary statistics used in affect research (besides mean and variance which are reproduced trivially): autocorrelation $\tau$, PA-NA correlation $\rho$ and the root-mean-square of successive differences (RMSSD). This was also done using a parametric bootstrap. A full overview of the results can be found in S2 Appendix. Here, we summarize the main results.

The AIM is able to reproduce the thee summary statistics. Its performance is comparable to that of the two Gaussian models. Only for reproducing positive PA-NA correlations, a major difference can be observed between the AIM and the two Gaussian models. The AIM cannot reproduce a positive PA-NA correlation, unless spuriously. Such positive correlations only appear in a small proportion of the data sets and are small when they do (indicating that they could be spurious). Similarly, negative autocorrelations are atypical for the three models. When they occur, they are spurious. Again, negative autocorrelations only appear in a small number of data sets and are small when they do.

## Cross-validation

Aside from focusing on how well the AIM and the two benchmark models are able to capture specific features, we also assessed to which extent the models differ in their overall predictive performance. By predictive performance, we mean that if a model is a good representation of the data, it should generalize well to new, previously unseen, data [48]. Practically, we tested the predictive performance using leave-one-out cross validation [49].

The results of the leave-one-out cross validation are summarized in Fig 11. The row and column labels refer to the models that are being compared. The histograms depict the differences in the average leave-one-out min-log-likelihoods $\Delta\bar{\ell}$ between the model indicated by the row label and the model indicated by the column label. If $\Delta\bar{\ell} < 0$, the predictive performance of the row model is better. Otherwise that of the column model is better. The percentage of data sets for which the column model has the better predictive performance is also indicated in the histograms.

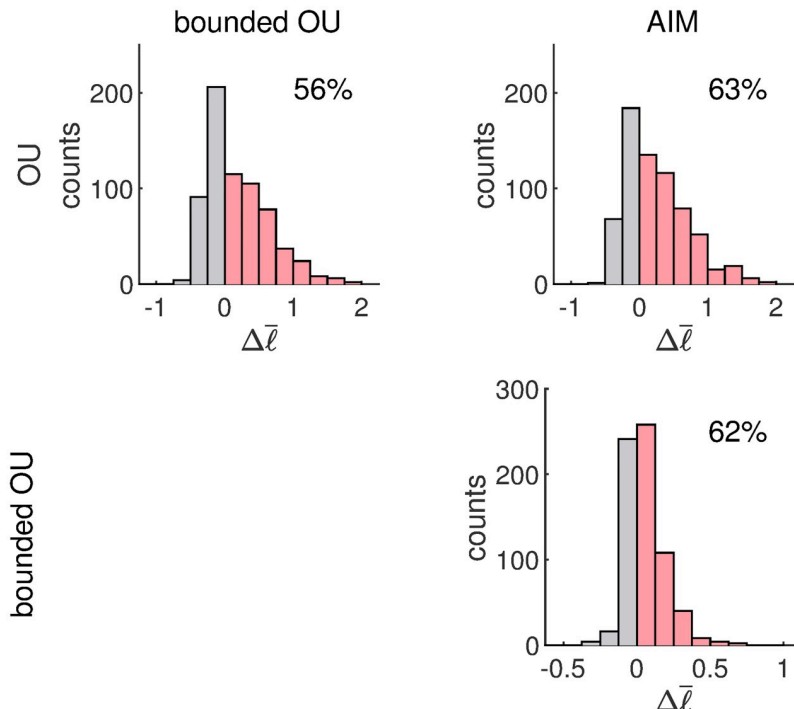

**Fig 11. Results of the leave-one-out cross validation.** The histograms depict the counts of the difference in average leave-one-out min-log-likelihood $\Delta\bar{\ell}$ of the model indicated by the row label and the model indicated by the column label. Whenever $\Delta\ell<0$, the model indicated by the row label has a better predictive performance and vice versa. The percentages correspond to the proportion of data sets for which the column model has a better predictive performance (data sets falling to the right of the origin).

According to the histogram in the top left corner of Fig 11, the predictive performance of the bounded OU model is better than that of the plain OU model in 56% of the cases. The $\Delta\bar{\ell}$ distribution is moreover heavily skewed towards the bounded OU model. Although the predictive performance of the plain OU model is better for 44% of the data sets, the difference in performance remains relatively small in comparison to when the bounded OU model outperforms the plain OU model. This can be understood by the fact that the bounded OU model is able to mimic the plain OU model but not the other way round. Whenever the observed data patterns have Gaussian features (as in Figs 6 and 8), the performance of the models is comparable. The heavy skew suggests however that there is a significant number of data sets with features that deviate from Gaussianity that are therefore better described by the bounded OU model (the results above indicate that the bounded OU model is better at accommodating non-Gaussian features). A similar reasoning holds for the comparison of the plain OU model and the AIM. The results are shown in the top right panel of Fig 11. The AIM has a better predictive performance than the plain OU model for 63% of the data sets.

In the bottom right panel of Fig 11 we see that the AIM moreover has a better predictive performance for 62% of the data sets when compared to the bounded OU model. The $\Delta\bar{\ell}$ distribution is still skewed in the direction of the AIM, but less than in the top right panel. The proportion of data sets for which the AIM has the better predictive performance however (more or less) remains the same as in the top right panel. Data sets with Gaussian patterns are well captured by models that are able to reproduce such features. While both the AIM and the

bounded OU model are capable of mimicking such features, the complexity of the bounded OU model is smaller. The right skew of the $\Delta\bar{\ell}$ distribution indicates however that the AIM is significantly better at dealing with non-Gaussian features.

In conclusion, the predictive performance of the AIM is better than that of both the plain OU model and the bounded OU model. For data sets with Gaussian features, the performance of the models is comparable. However, the significant number of data sets for which the AIM has a better predictive performance suggests that non-Gaussian features are rather common. This is in line with the results depicted in Fig 5 (data patterns with large centralized third moments), Fig 7 (data patterns with large curvatures) and Fig 9 (data patterns with large multimodality coefficients).

## Discussion

This paper introduced the Affective Ising Model (AIM) as a computational model for the dynamics of positive affect (PA) and negative affect (NA) across time. It has been derived from abstract microscopic (neurophysiological) principles: two pools of positive and negative binary, stochastic information units (binary neurons) continuously act upon one another. Within a pool, the binary neurons excite each other and between pools they inhibit one another. Each binary neuron is additionally characterized by a threshold that needs to be surpassed in order to become active. External influences act on the binary neurons; they can activate them or impede their activity.

These binary neurons can be thought of as all the elements determining the affect state of an individual. As such, they are related to all neuronal, physiological and psychological processes that play a role in the affect system. Since this entire system is extremely complex it defies a detailed, encompassing mathematical description. Therefore, inspired by concepts of statistical mechanics, the AIM averages across all these separate microscopic contributions and shifts the attention to the bivariate affect state that is composed of the average activity of the positive and negative pools respectively, denoted as PA and NA. This bivariate affect state is continuously subject to change. As a consequence, the AIM gives rise to a continuous-time bivariate system of stochastic differential equations that describes how the affect state fluctuates across time. In the long time limit, the probability of affect states attainable by the system relaxes to an equilibrium. Like the dynamics, this equilibrium distribution is person specific, reflecting individual differences.

The results indicated that the AIM is able to explain signature nonlinear empirical phenomena, such as the observed skewness of affect distributions, the V-shaped relation between PA and NA, and bimodality. They also indicated that Gaussian models (a plain OU model and a bounded version thereof) closely related to the vector autoregressive models that are typically used in psychology are not capable of explaining these phenomena to a satisfactory degree. Furthermore, the AIM incorporates statistical measures, like the autocorrelation, the PA-NA correlation and the root-mean-square of successive differences, as well as the Gaussian models. Such summary statistics are often used to extract information about important features of the data. However, they do not allow us to make predictions; they cannot be used to predict what the outcome will be at the time of the next measurement or what will happen when a specific stimulus is given, because they do not provide us with information on how the affect system updates across time. A model-based approach like the AIM is therefore preferred, but it is important that meaningful statistical measures are preserved.

In conclusion, the AIM seems to be a good model to support the data. It nicely integrates the statistical measures, is capable of reproducing the observed non-Gaussian features and does not show signs of major misfit beyond that of the Gaussian models.

## Limitations and further developments

The data analyzed in this paper have been collected under a specific protocol, involving decisions regarding the emotion items that were included in the questionnaires, the response format (continuous sliders), labels that were used, the specific formulation of questions and the sampling rate. The results described in this paper concerning features such as bimodality and skewness are likely to be influenced by these choices. In this paper, we did not wish to make any strong substantial claims about the features themselves. We merely pointed out that features like skewness, V-shapes and bimodalities have been repeatedly reported in data, and that they were also common in the data included here.

The goal of this paper was mainly to introduce a model that is able to unify non-Gaussian features in a single dynamical framework. To show that the AIM is a viable model for affect dynamics, it was applied to real affect data. To do this, a method had to be introduced for linking the ESM measurements to the average activities of the two pools of stochastic binary neurons. In this paper, we opted for a measurement model that connects the average activities of the two pools to the respective averages of reported scores on the positive and negative emotion items.

The measurement model we used is a very simple one that involves no parameters. Because of the averaging over emotion items, we also considered measurement noise to be integrated out. The model could nevertheless be complemented with more elaborate measurement models that explicitly take into account measurement noise, using item response theory for example [50]. This would ensure that the model is more widely applicable to different experimental protocols (e.g., using Likert scales instead of continuous scales, using other emotion items, or using different labels). An appropriate measurement model will be essential if one seeks to validate or falsify certain substantial claims concerning affect dynamics.

In this paper, we have disregarded input to the affect system because objective information concerning context was hardly available in the data that were analyzed. The AIM nonetheless provides a natural framework to account for external disturbances but to do so, there are still some important aspects to be considered. One of the major issues with contextual disturbances in ESM studies is that we generally have no access to the entire time course of external disturbances, we only get fragmented information. This makes it difficult to assess which stimuli exactly influence a measurement. In between measurements, several things may have happened to the individual. At the moment of observation, we observe the accumulated effect of everything that has happened. In general, we have no exact idea of what happened when, let alone that we are informed of everything that has happened.

To study the influence of stimuli on affect, it may make more sense to use experiments (e.g., watching a film fragment or listening to music). Such stimuli can be coded, making them easier to model and providing continuous information of what is happening. Such coded stimuli can be approximated and implemented using, for instance, boxcar functions (i.e., constant nonzero during a specific time interval and zero elsewhere). Furthermore, experiments have the advantage that different participants can be given exactly the same input, allowing us to study individual differences. Participants can be given specific input to artificially disrupt their affect system and their responses can be compared to predictions made by a model such as the AIM. Controlled experiments which involve large and objectively quantifiable input are therefore a good way to validate affect models.

The AIM presented in this paper included only two distinct pools (PA and NA). On the one hand, this was a consequence of theoretical considerations—PA and NA are the principal dimensions of affect. On the other hand, computational limitations hamper the extension of the AIM to more dimensions. Lifting these limitations could greatly extend the scope of the

model's applicability. It has been established that, at the most basic level, affective experiences are for a large part accounted for by a two-dimensional space [19, 33]. Some authors have proposed up to four principal dimensions however [51]. In case it turns out that more than two dimensions are needed to optimally characterize affective experiences, we may consider an extension of the AIM with additional pools of binary neurons. Although the computational technique developed for this paper is sufficiently fast and reliable, it suffers tremendously under the curse of dimensionality. If this limitation were removed, a multidimensional model would be tractable.

We did not attach strong—psychological or neurophysiological—interpretations to the parameters of the AIM. Instead we focused on distributional patterns and qualitative features. The reason is that parameter interpretation, although appealing, is very difficult because of the limited number of applications of the AIM at this point. A simple parameter transformation could give rise to a new set of parameters, with different interpretations, but with exactly the same ability to explain the data. By setting up experiments to establish selective influence (i.e., manipulations targeting only one parameter), we might be able to gain more information on possible interpretations.

The AIM can accommodate a wide variety of distributional patterns in a two-dimensional affect space, among which distributions that are heavily skewed, exhibit V-like patterns, or that are bimodal. The spectrum of distributional patterns that may emerge from the AIM is however much broader. These features may help us make predictions about outcomes of experimental manipulations. Below we give two examples of such predictions: mixed emotions and phase transitions.

First, in a two-dimensional space, AIM distributions can have up to four modi (or homebases). The locations of these homebases are always roughly situated in the four quadrants of the unit square. They can be pressed close together or be pushed far apart. Inhibition typically removes the homebase in the quadrant corresponding to highly activated states (high PA, high NA). However, although the AIM's inhibition parameter suppresses the occurrence of such mixed states, they are not impossible. Such highly activated states have been recorded in real life and are labeled as mixed emotions [52]. Using model fits, one could make predictions about which persons are more likely to experience mixed states and which persons are less likely to experience them.

Second, due to its nonlinearity, the AIM (like other nonlinear dynamical models) can incorporate phase transitions (e.g., [53]). Around a phase transition, an emotional landscape corresponding to that of a healthy individual could give rise to a stable homebase in the high NA region with the slightest perturbation of the AIM's parameters. This could have implications for understanding, for instance, proneness to depression or other mood disorders.

Exploring the novel predictions that result from the AIM will be an important task. Such novel predictions have the advantage that they may be able to falsify the model or corroborate it.

## Materials and methods

### Data

In this paper, data sets from two different studies were analyzed. All data have been obtained using the Experience Sampling Method (ESM) in which individuals are repeatedly asked to report on their momentary affect states in the context of daily life using smartphones. They both complied with local ethical regulations and were approved by an institutional ethics committee. All participants provided informed consent. Only the aspects of the data collection relevant for our study will be described below.

**Study 1: Three-wave ESM study.** In the first study, data were collected in an attempt to assess real-life affect dynamics of students throughout their freshman year of college. An elaborate description of it is given in [54]; it was approved by the institutional review board of the KU Leuven (ML8514-S54567).

Students were recruited to reflect a broad range of psychological well-being (in terms of severity of depressive symptom). The study consisted of three independent assessment periods or measurement waves in which the students took part in a one-week ESM protocol. The first ESM wave coincided with the beginning of the academic year. Wave 2 and Wave 3 followed respectively four and twelve months later. A remuneration was provided for participation; €60 for each completed wave and an additional €60 for completing all waves. Initially, 180 participants were selected from a pool of 686 students (235 males) starting freshman year. After the study had already begun, another 22 participants were recruited to ensure the desired sample size. This led to an initial sample of 202 participants. In the first wave, there were two participants who showed a remarkably low compliance (a response rate below 50%). Excluding those, the initial sample consisted of 200 participants (55% female; $M_{age}$ = 18 years; $SD_{age}$ = 1 year). Due to dropout, 190 participants (56% female) remained in Wave 2, and 177 participants (56% female) remained in Wave 3.

In the analyses presented in this paper, data sets from the same participant but from different waves were treated separately, resulting in a total of 567 data sets for this study.

**The ESM protocol.** In each measurement wave, momentary affect was assessed using ESM. The participants were given a Motorola Defy Plus Smartphone which they had to carry throughout their normal daily activities. The smartphones were programmed to beep 10 times a day in between 10 a.m. and 10 p.m. for seven consecutive days. A stratified random interval scheme was used to sample the beep times; a beep occurred on average every 72 minutes. Each beep prompted the participants to rate their positive (*happy*, *relaxed*, *cheerful*) and negative (*sad*, *anxious*, *depressed*, *anger*, *stressed*) emotions by means of a continuous slider ranging from 0 (*not at all*) to 100 (*very much*). On average, participants responded to 87% (*SD* = 9%), 88% (*SD* = 9%) and 88% (*SD* = 9%) of the beeps for the three waves respectively.

**PA and NA scores.** PA and NA scale scores were constructed for each beep by averaging the responses to the individual positive and negative emotion items respectively. The scores were then re-scaled to lie within the unit interval, as required for fitting the AIM (because the affect state variables $y_1$ and $y_2$ are proportions of active stochastic binary elements).

**Study 2: Clinical ESM study.** The second study was set up to study symptom and emotion dynamics in individuals suffering from major depressive disorder and/or borderline personality disorder, and healthy controls. A more elaborate description of this study is given in [55]; it was approved by the Medical Ethics Committee UZ Leuven (B322201627414).

A clinician screened patients in three Belgian psychiatric wards (*KU Leuven hospital UPC Sint-Anna*; *UPC De Weg/Onderweg*; and *Broeders Alexianen Tienen hospital ward Prisma II*). Patients who were deemed eligible for enrollment in the study were interviewed by a clinically trained researcher using the Dutch version of the Structured Clinical Interview for DSM axis I disorders (SCID-I; [56]) and the Borderline Personality Disorder (BPD) subscale of the DSM axis II disorders (SCID-II; [57]). If patients met the criteria for one of the mood or personality disorders, they were included. If they were, however, acutely psychotic, acutely manic, addicted, or diagnosed with a (neuro)cognitive disorder at the time of the interview, they were excluded.

A sample of healthy controls was recruited via advertisements, social media, flyers and by the Experimental Management System of the KU Leuven (university of Leuven). Their age and gender distribution were made to match those of the clinical patients. From the initial sample, some participants (clinical patients as well as healthy controls) were removed because

of complications during the ESM protocol or because of a low compliance. The final sample consisted of 118 participants of whom 38 were diagnosed with major depressive disorder (50% female; $M_{age}$ = 41 years; $SD_{age}$ = 14 years), 20 with borderline personality disorder (95% female; $M_{age}$ = 30 years; $SD_{age}$ = 12 years), another 20 with both (85% female; $M_{age}$ = 33 years; $SD_{age}$ = 11 years), and a control group without current psychiatric diagnosis consisting of 40 participants (58% female; $M_{age}$ = 35 years; $SD_{age}$ = 12 years).

The participants were compensated after the ESM assessment. Those who had a compliance rate above 75% received €35. For every ten percent that the compliance rate was below this threshold, 5 euros were deducted.

**The ESM protocol.**    The measurement protocol was similar to the one used for the previous study. Motorola Defy Plus Smartphones were given to the participants and had to be carried with them throughout their daily lives. They were programmed to beep 10 times a day in between 10 a.m. and 10 p.m. for seven consecutive days. A stratified random interval scheme was used to sample the beep times; a beep occurred on average every 72 minutes. Each beep prompted the participants to fill out a questionnaire which consisted of 27 questions, including questions about emotions, social expectancy, emotion regulation, context and psychiatric symptoms. It took participants on average 2′2″ ($SD$ = 37″) to fill out a questionnaire. The positive (*euphoric*, *happy*, *relaxed*) and negative (*depressed*, *stressed*, *anxious*, *angry*) emotion items could be rated using a continuous slider ranging from 0 (*not at all*) to 100 (*very much*). On average, the participants responded to 87% ($SD$ = 11%) of the beeps.

**PA and NA scores.**    As for Study 1, the PA and NA scale scores were calculated as average item responses and re-scaled to the unit interval.

## Fitting the models

The AIM was fitted using a maximum likelihood optimization procedure. Because of the nonlinearity of the model (see logarithmic terms in the free energy function in Eq (2)) there is no analytic expression for the likelihood nor for the maximum likelihood estimates. Simulation-based techniques were therefore used to compute the likelihoods, and the differential evolution global optimization heuristic [58] was relied on to find the maximum likelihood estimates. For a more elaborate description of how the AIM was fitted, see S1 Appendix.

Besides the AIM, two other models were considered in this paper for reasons of comparison. The first of these models was a continuous-time vector autoregressive model, also known as the Ornstein-Uhlenbeck (OU) model [21, 45]. It is inherently Gaussian and therefore served as a benchmark. One of the issues with the plain OU model, however, is that it generates normal distributions that are unbounded while the measurement is constructed to lie in a unit square. Thus, the plain OU model assigns probability mass outside the region of support of the measurements. To account for this, we also considered a truncated variant of the OU model in which hard boundaries at the edges of the unit square were introduced. This bounded OU model was also used to analyze the data. This model is not commonly encountered, but it serves as a good model intermediate between the OU model and the AIM. It generates Gaussian distributions that are truncated at the boundaries and as a result, the generated distributions are still unimodal but can display skew. The maximum likelihood estimates of the plain OU model were computed using an analytical min-log-likelihood function, those of the bounded OU model were computed using the same numerical method as the one used for the AIM.

**Evaluating the models.**    In this section we discuss how the AIM and the two benchmark models were evaluated. There are three sets of evaluation criteria. First, we assessed to which extent the models can explain the non-Gaussian features that are present in the data (skew, V-

shape and bimodality). It was expected that the plain OU model would fail, while the bounded OU might be able to deal with skew and possibly also V-shape, but not multimodality. Second, we evaluated whether the models can encompass a number of commonly used summary statistics to describe the affective dynamics (autocorrelation, root-mean-square successive differences, correlation between PA and NA). Third, we compared the models regarding their predictive accuracy.

**Explaining the nonlinear empirical features.** To analyze to what extent the AIM is able to systematically capture the empirical features, we defined for each of them a statistic, which were estimated using a parametric bootstrap (see e.g., [47]). In a first step, the statistics were computed on every one of the 685 observed data sets. Second, the model was fitted to each data set and using the maximum likelihood estimates, 1,000 new (replicated) data sets were generated. The fitted model was any of the three models mentioned above (plain OU, bounded OU and AIM). For more information on how data was simulated, see S1 Appendix. Each replicated data set had the same sample size as the original data set. Third, the statistics were computed for each of the replicated data. Fourth, the distributions of the replicated statistics were used to gauge the plausibility of the observed statistic.

Below follows an overview of the features and the corresponding statistics that were defined.

**Skewness.** The skewness was assessed by means of the standardized third moment $m_3$. For simplicity, the skewness of PA and that of NA were treated separately. This should not be a major concern since signs of skewness have been reported for both dimensions [14].

**V-shape.** The inhibition between the PA and NA dimension results in a region where it is unlikely to find any observations; as can be seen in Fig 12(a), the upper right region corresponding to highly excited PA and NA pools shows little to no observations. Regions where the excitation of at least one pool is small get frequented more often. This can lead to a boomerang shape or a V-shape (when rotated counterclockwise over an angle of 45˚).

The curvature $\kappa$ of the kink of the V-shaped data pattern was analyzed by means of a parametric bootstrap. This curvature $\kappa$ was operationalized as the largest curvature of the least-squares parabola within the data domain. Specifically, the least-squares parabola was first of all fitted on the data. Prior to fitting the parabola, the affect space was rotated counterclockwise over an angle of 45˚; afterwards, the affect space was rotated back to its original position.

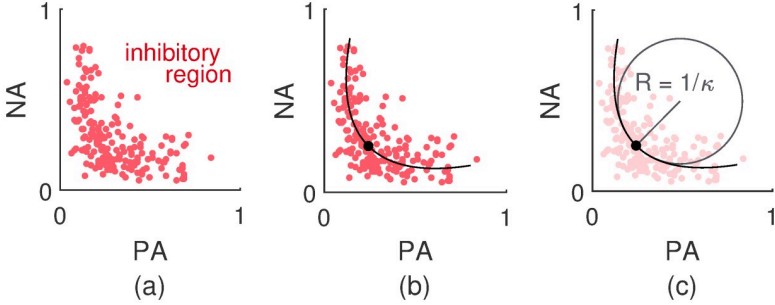

**Fig 12. Visualization of how the V-shaped data patterns were analyzed.** In panel (a) an example of a V-shaped data pattern is drawn. The inhibitory region corresponds to a region in the affect space were both dimensions inhibit one another, making it hard for the affect state to actually reach there. Regions where the excitation of at least one of the pools of binary neurons is small are more accessible. This gives rise to a V-shaped, curved data pattern. In Panel (b), the least-squares parabola through the data points of Panel (a) is shown. The point with the largest curvature is indicated as well. In this case, this point coincides with the vertex of the parabola. In Panel (c), the least-squares parabola from Panel (b) is drawn again and the osculating circle at the point with the largest curvature is made explicit. The inverse of the radius $R$ of this circle is defined as the curvature $\kappa$.

In Fig 12(b), an example of a least-squares parabola obtained in this manner is shown. The point with the largest curvature $\kappa$ is indicated as well. This is the point of the parabola for which the corresponding osculating circle has the smallest radius $R$, since curvature is defined as the inverse of the radius of the osculating circle. In Fig 12(c), the osculating circle of the parabola at the indicated point is drawn. In general, the point with the largest curvature $\kappa$ coincides with the vertex of the parabola but this can deviate because the parabola is limited to the data domain. Curvatures corresponding to upward parabolas are positively defined. Parabolas pointing downwards have a negative curvature.

The V-shapes of interest are those that stretch out over and are supported by larger areas, but compact data patterns can naturally accommodate larger curvatures. To diminish undesirable fortuitous curvatures, the data have been standardized before applying the procedure.

From a theoretical perspective it makes little sense to allow for an excitatory interaction between the PA and NA dimensions, rather than inhibition. Therefore, the parameter $\Lambda_{12}$ was constrained to be larger than or equal to zero in all our optimizations. In doing so, however, the number of possible affect distributions was constrained and with it the possible V-shaped patterns. To see whether the rationale behind inhibition between the PA and NA dimensions truly holds, the effect of liberating the interaction parameter $\Lambda_{12}$ was also investigated.

**Bimodality.** Bimodality was assessed using the bimodality coefficient ($BC$; [59]). In terms of the sample size $n$, the skewness $m_3$ and the excess kurtosis $m_4$, it is defined as:

$$BC = \frac{m_3^2 + 1}{m_4 + 3\frac{(n-1)^2}{(n-2)(n-3)}},$$

where both the skewness $m_3$ and the excess kurtosis $m_4$ are corrected for sample size. It has a value between 0 and 1. Larger values point toward bimodality, smaller values point toward unimodality. Typically, the $BC$ of an empirical distribution is compared to a benchmark value of $BC_{crit} = 5/9 \approx 0.5556$; this is the value that is expected for a uniform distribution.

Because the $BC$ is only defined for univariate distributions, the marginalized PA distribution and the marginalized NA distribution were considered (similar to skewness). To ensure that no bimodality goes unnoticed, the marginalized PA+ NA and PA–NA distributions (the distributions obtained by projecting on the diagonals of the affect grid) were considered as well. The $BC$ was computed for all four marginal distributions and the largest value was used.

Although bimodality coefficients can give an indication of bimodality, large $BC$ values can also occur for heavily skewed distributions [59]. In other words, large $BC$ values are no guarantee for multiple modes. Therefore, we also investigated in how many cases the maximum likelihood estimates of the AIM corresponded to a multimodal distribution.

To take uncertainty on the estimates for the $BC$ estimates into account, a non-parametric bootstrap was used. We replicated the entire population of 685 data sets 200 times. For each replication, we re-sampled every data set by drawing as many observations from the original data set as there are in the original data set, with replacement. We then computed the maximum likelihood estimates of each re-sampled data set and the percentage of bimodal distributions in every population replication. Ultimately, we ended up with a distribution of 200 percentages.

## Reproducing common summary statistics

The models were also evaluated regarding how well they are able to describe three of the more common summary statistics used in affect research (besides mean and variance which are reproduced trivially): autocorrelation, PA-NA correlation and the root-mean-square of

successive differences (RMSSD). The autocorrelation and RMSSD explicitly take into account the time dynamics, whereas the PA-NA correlation is more related to the shape of the distributions. The same parametric bootstrap procedure was used as for the evaluation of the three nonlinear features.

Because we expect any reasonable model for affect to be able to capture these linear, Gaussian features, we have relegated a full discussion of the analysis method and results to the S2 Appendix.

**Cross-validation.** We also assessed to which extent the models differ in their overall predictive performance using leave-one-out cross validation [49]. This is a procedure in which each observation of a data set is left out once and is predicted using the maximum likelihood estimate of the model obtained using the remainder of the observations. The likelihood of the true observation given the model prediction is then a measure of the predictive performance of the model.

In practice, the predicted min-log-likelihood $\ell$ is computed, which is the negative logarithm of the predicted likelihood and a measure of badness-of-prediction. For each data set, we computed the average $\bar{\ell}$ of the predicted min-log-likelihoods. The smaller the average predicted min-log-likelihood, the better the predictive performance of the model.

## Supporting information

**S1 Appendix. Fitting the AIM.** In this appendix, the numerical procedure that was used to fit the AIM is explained in detail, which is similar to a procedure used by [60].
(PDF)

**S2 Appendix. Reproducing common summary statistics.** In this appendix, the method for analyzing the three summary statistics, autocorrelation, PA-NA correlation [61] and the root-mean-square of successive differences [4], is described. The results of the analysis are discussed as well.
(PDF)

## Author Contributions

**Conceptualization:** Tim Loossens, Merijn Mestdagh, Francis Tuerlinckx, Stijn Verdonck.

**Data curation:** Tim Loossens.

**Formal analysis:** Tim Loossens.

**Methodology:** Tim Loossens, Stijn Verdonck.

**Resources:** Egon Dejonckheere.

**Software:** Tim Loossens.

**Supervision:** Francis Tuerlinckx, Stijn Verdonck.

**Validation:** Merijn Mestdagh, Francis Tuerlinckx, Stijn Verdonck.

**Visualization:** Tim Loossens.

**Writing – original draft:** Tim Loossens, Egon Dejonckheere, Peter Kuppens, Francis Tuerlinckx, Stijn Verdonck.

**Writing – review & editing:** Tim Loossens.

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
