## [Decision Letter · Decision Letter 0]

8 Jan 2020

Dear Dr Loossens,

Thank you very much for submitting your manuscript 'The Affective Ising Model: a Computational Account of Human Affect Dynamics' for review by PLOS Computational Biology. Your manuscript has been fully evaluated by the PLOS Computational Biology editorial team and in this case also by independent peer reviewers. The reviewers appreciated the attention to an important problem, but raised some substantial concerns about the manuscript as it currently stands. While your manuscript cannot be accepted in its present form, we are willing to consider a revised version in which the issues raised by the reviewers have been adequately addressed. We cannot, of course, promise publication at that time.

Sincerely,

Jacopo Grilli

Associate Editor

PLOS Computational Biology

Stefano Allesina

Deputy Editor

PLOS Computational Biology

[LINK]

Reviewer's Responses to Questions

**Comments to the Authors:**

Reviewer #1: The Affective Ising Model: a Computational Account of Human Affect Dynamics (PCOMPBIOL-D-19-01513 – Review)

This article proposes a nonlinear stochastic model for studying the dynamics of affect in humans, labeled by the authors the Affective Ising Model (AIM). The model is based on neurophysiological and behavioral fundamentals underlying positive and negative affect, and is a model that can account for various types of manifestations of affect such as skewness, multimodality, and non-linear relations of positive and negative affect. As part of their illustration, the authors implement the model to two empirical data sets of affect from experience sampling studies, and examine the extent to which it can reproduce the various manifestations of affect, comparing it with two vector AR models fitted in continuous time. The findings indicate that the AIM model is better at mapping onto manifestations of affect data (especially non-Gaussian manifestations) than the AR models, and it has a higher predictive value for the time series of affect.

I liked this paper very much. I think it can make a very important contribution to our theoretical understanding of affect dynamics. My main concern has to do with the format. In its current form, the paper is a bit overwhelming. It is excessively long, with a very large number of figures (19 currently). Some sections read more like a technical report, whereas other sections (i.e., implementation of the model) are more manuscript-type, and very engaging. A full description of the model and its underpinnings needs space, and at times it feels like part of the supplementary material should be included in the main paper as well. On the other hand, combining this explanation and the technical details with the implementation of the model to the two data sets, makes the paper a bit unwieldly. I wonder if it would make sense to split the manuscript into two papers. One could be a more technical paper with all the necessary details, including those from the appendix about fitting the model. The other paper could focus on the implementation of the model to the empirical data sets, comparing the results with those from the two AR models, and fleshing out the implications in terms of affect dynamics.

Reviewer #2: In “The Affective Ising Model: A computational account of human affect dynamics”, the authors present a novel computational model for describing and predicting dynamics in self-reported affect. In the process of introducing their model, they analyze two sets of experience sampling data in order to demonstrate the ability of the Affective Ising Model (AIM) to accurately capture observed self-reported affect dynamics and compare the AIM’s performance to other widely used methods of modeling such data. Overall, the manuscript is well-written and makes an interesting and important contribution to the affective sciences. However, there are a number of issues that I believe deserve greater consideration and clarification. The authors should provide additional explanation/resources to aid other researchers in the practical implementation of their model. Additionally, the current manuscript does not adequately address how the model and its parameters should be understood and interpreted in relation to other work or what findings might suggest either future revision or falsification. Last, there remain questions of the model’s predictive abilities, stemming from the fact that the model is likely most properly understood as a model of dynamics in self-reported affect. I elaborate each of these points below. Overall, I am enthusiastic about the work that the authors have done and believe that efforts to incorporate computational modeling to dynamics in experience sampling such as this will be critical for moving work in this area forward.

The authors do a noteworthy and commendable job of describing their model is a way that facilitates an intuitive understanding for readers. In fact, it’s so intuitively described that it’s almost easy to overlook the challenges of practical implementation. To enable other researchers (especially those who are newer to these sorts of models) to use and critique novel computational models, it is important for authors to provide access to materials such as code. This is especially useful because many researchers find actual code as the clearest means for learning the ins and outs of a model, aside from the more formal mathematical description. Therefore, I request that the authors provide access to code for their model. While the full code would likely be the most helpful, a stripped-down version and/or pseudocode, if possible, (e.g., as in Turner et al., 2013, Psychological Methods) could also be helpful. Additionally, a supplement detailing practical challenges, requirements, and limitations for implementing the model would be beneficial. Presenting a model such as the AIM without discussing any of the practical aspects will limit its usage among other research groups.

On the topic of clear explication of their model, it was not entirely clear to me how inputs to the model are handled. The authors explicitly state that they avoid discussing inputs in this manuscript, but I believe a discussion of how this might be done warrants greater discussion. Could the authors elaborate on how external events might be fed into the model? Can they provide some proposals for how researchers might code external events? The discussion of this (lines 817-822) could be developed a bit further.

The authors spend a significant portion of the Model Description section justifying the AIM as a computational model that seeks to describe macroscopic phenomena and not the “microscopic” component processes. To help clarify the objectives of their model to readers, it would likely be useful for the authors to draw on already established frameworks such as Marr’s (1982) levels of analysis, as has been done elsewhere in the emotions literature (e.g., Adolphs & Andler, 2018; Bach & Dayan, 2017). Using Marr’s labels, it seems then that the AIM is meant for an algorithmic level of analysis as it does not appear to make claims about either the purpose of the algorithm or its neural implementation.

Related to the question of levels of analysis is the question of how the authors believe their model might be informed by research at other levels of analysis. Given the emphasis on the fact that the AIM is intended to capture macroscopic phenomena and not their microscopic components, it would be useful to know whether or not the authors expect research at other levels of analysis to inform, modify, or constrain their model in the future. This question ties into a broader question about falsifiability. For example, are there “affective landscapes” which the AIM cannot produce? (e.g., tri-modal?) As the authors show, constraining the values of the “inhibition” parameter rules out landscapes where PA and NA are positively correlated. However, the participants in their own dataset who showed a positive correlation and other research (e.g., arousal-focused individuals; Barrett & Bliss-Moreau, 2009) would seem to suggest that this parameter constraint should eventually be relaxed.

The authors do not provide much in the way of discussion of possible psychological interpretations of the model parameters. Perhaps this is because the authors expect that there are no direct concept to parameter mappings, only indirect (e.g., Roberts & Hutcherson, 2019). If this is the case, the authors should be explicit about the reasons for their reticence to speculate about how their model’s parameters connect to psychological concepts so that readers can be directed to use the same caution. Do the authors believe that no psychological interpretations should be given to the model parameters and that only features of the resulting “affective landscape” should be given psychological meaning (e.g., the possibility of multiple homebases in patients with borderline personality disorder mentioned in lines 842-847)? The manuscript should offer greater clarity about how the model might be used and interpreted as any readers will certainly want to assign psychological meaning to the parameters.

The authors compare their model to two other models that are Gaussian, with or without bounds. One of the highlighted strengths of the AIM is its ability to deal with the boundedness of the data and produce the observed skewness. However, this demonstration raises the unanswered question of to what degree these aspects of the data are due to affective dynamics versus due to how people use the sorts of self-report scales they are provided. To be clear, I do not doubt that affective experience is non-normally distributed. However, it seems highly likely that some of the distributional features of the data are also contributed to by the self-report measures themselves. For example, though participants may use the most extreme response for negative affect on an occasion, is this capturing their most extreme possible level of negative affect? Similarly, though participants appear to frequently report no negative or positive affect at all, is this truly their “flattest” affective experience? All of this suggests the subtle, yet important, distinction between a model of affective dynamics versus a model of dynamics in self-reported affect. I do not see this point as an invalidation of the AIM, but realizing this distinction raises important additional questions. For example, if the AIM is a model of dynamics in self-reported affect, how might a person’s “affective landscape” be influenced by changes to the self-report measure (e.g., scale anchors, items, etc.)? Will fitting the AIM to an individual’s data collected using one self-report measure allow for accurate predictions of data collected using a different measure? If there are discrepancies in how the model fits to data derived from different measures, how should these be dealt with? It would seem important for the authors to acknowledge these questions and how they might be addressed.

A couple other minor points are presented below:

On lines 103-106, the authors state “Most likely the biological structures dealing with positive and negative affect information consist of a large number of smaller elements (neurons, neurotransmitters, hormones, etc.) that in interaction with psychological processes produce positive and negative feelings.” I acknowledge that my objection here is possibly a bit idiosyncratic, but I always have trouble understanding what is meant by a sentence such as this. Presumably the authors agree that biological structures such as neurons, neurotransmitters, and hormones are part of the material substrates for psychological processes. So what is meant by neurons interacting with psychological processes? To me this sounds like a subset of elements interacting with their superset, which does not make sense to me. Do the authors mean to say something about how psychological processes are constrained by aspects of their biological underpinnings? Or how features of the biological structures interact with an organism’s broader ecology and history?

The sentence on lines 189-191 appears to be missing a “the”: “As a result, absorbing such an input process into THE model does not affect the model’s 190 properties (e.g., structure, dynamics, distributions, etc.).”

**Have all data underlying the figures and results presented in the manuscript been provided?**

Reviewer #1: Yes

Reviewer #2: Yes

PLOS authors have the option to publish the peer review history of their article (what does this mean?). If published, this will include your full peer review and any attached files.

Reviewer #1: No

Reviewer #2: No

---

## [Decision Letter · Decision Letter 1]

8 Apr 2020

Dear Mr Loossens,

We are pleased to inform you that your manuscript 'The Affective Ising Model: a Computational Account of Human Affect Dynamics' has been provisionally accepted for publication in PLOS Computational Biology.

Best regards,

Jacopo Grilli

Associate Editor

PLOS Computational Biology

Stefano Allesina

Deputy Editor

PLOS Computational Biology

Reviewer's Responses to Questions

**Comments to the Authors:**

Reviewer #2: The authors’ revisions have greatly enhanced the overall clarity of the manuscript and helped to highlight key questions that will need to be addressed in future work. I have no further questions or concerns to raise at this point. I recommend acceptance of this manuscript for publication.

**Have all data underlying the figures and results presented in the manuscript been provided?**

Reviewer #2: None

PLOS authors have the option to publish the peer review history of their article (what does this mean?). If published, this will include your full peer review and any attached files.

Reviewer #2: No

---

## [Editor Report · Acceptance letter]

1 May 2020

PCOMPBIOL-D-19-01513R1 

The Affective Ising Model: a Computational Account of Human Affect Dynamics

Dear Dr Loossens,

I am pleased to inform you that your manuscript has been formally accepted for publication in PLOS Computational Biology. Your manuscript is now with our production department and you will be notified of the publication date in due course.

With kind regards,

Laura Mallard
